# Probing proton PDFs at high $x$ with LHCb

**Thomas Boettcher$^\star$ on behalf of the LHCb Collaboration**

University of Cincinnati, Cincinnati, OH, USA

$\star$ boettcts@ucmail.uc.edu

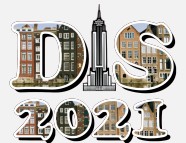

*Proceedings for the XXVIII International Workshop
on Deep-Inelastic Scattering and Related Subjects,
Stony Brook University, New York, USA, 12-16 April 2021*

## Abstract

**LHCb is a forward spectrometer at the LHC covering the pseudorapidity region $2 < \eta < 5$. Because of this forward coverage, LHCb can probe the proton parton distribution functions (PDFs) in previously unexplored kinematic regimes, in particular at very high and low Bjorken-$x$. This contribution presents LHCb measurements that can be used to constrain the proton PDFs, with a focus on the high-$x$ and high-$Q^2$ regime.**

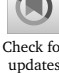

## 1 Introduction

The LHCb detector is a single-arm forward spectrometer at the LHC designed to study decays of hadrons containing $b$ or $c$ quarks [1]. LHCb is fully instrumented in the region $2 < \eta < 5$. As a result of this forward acceptance, LHCb can probe the proton parton distribution functions (PDFs) in kinematic regimes complementary to those accessible at central pseudorapidities. LHCb studies hard interactions between partons carrying large fractions of the proton momentum (high-$x$) and partons with small fractions of the proton momentum (low-$x$). Measurements of electroweak boson production at LHCb have been used to constrain the quark PDFs at high $x$ and high momentum transfer ($Q^2$). Futhermore, LHCb measurements of $Z$ + jet and $W$ + jet production, as well as measurements using identified $c$- and $b$-quark jets, provide additional information about the quark PDFs at high-$x$.

## 2 Forward $W$ and $Z$ production

LHCb has measured $W$ and $Z$ boson production using data from Run 1 of the LHC. The $W$ boson cross section has been measured at $\sqrt{s} = 7$ TeV using the $W \to \mu\nu$ channel [2] and at $\sqrt{s} = 8$ TeV using both the $W \to \mu\nu$ [3] and $W \to e\nu$ channels [4]. $W$ boson yields are extracted using fits to the charged lepton $p_{\rm T}$ spectra. The cross section is measured differentially in lepton pseudorapidity ($\eta_l$). The $Z$ boson cross section has been measured in both the

$Z \to \mu^+\mu^-$ and $Z \to e^+e^-$ channels at $\sqrt{s} = 7$ and $8\,\text{TeV}$ [3, 5–7]. The cross section is measured differentially in the rapidity of the $Z$ boson ($y_Z$). The measured $W$ and $Z$ differential cross sections at $\sqrt{s} = 8\,\text{TeV}$ in the muon channels are shown in Fig. 1 and are compared to next-to-next-to-leading-order (NNLO) calculations. LHCb has also measured $Z$ boson production at $\sqrt{s} = 13\,\text{TeV}$ in the $Z \to \mu^+\mu^-$ and $Z \to e^+e^-$ channels [8]. Additionally, LHCb has measured the $Z$ boson cross sections at $\sqrt{s} = 7$ and $8\,\text{TeV}$ in the $Z \to \tau^+\tau^-$ channel [9, 10].

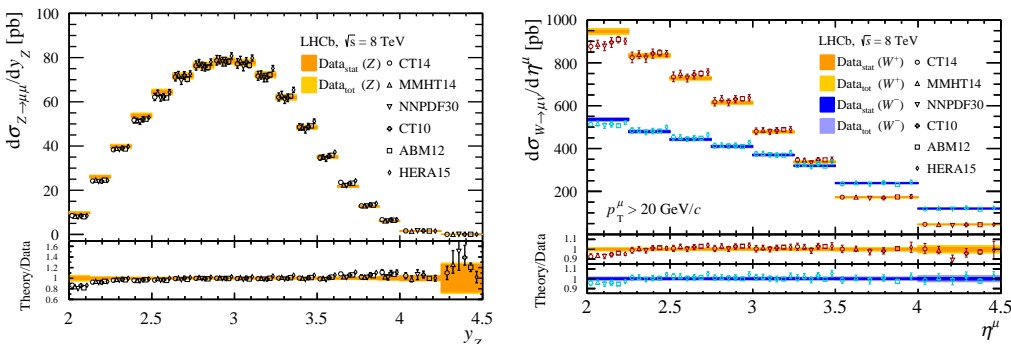

Figure 1: Measured $Z$ (left) and $W$ (right) boson differential cross sections at $\sqrt{s} = 8\,\text{TeV}$ [3].

LHCb Run 1 measurements of electroweak boson production have been used in the state-of-the-art CT18 [11], NNPDF3.1 [12], and MSHT20 [13] global PDF fits. LHCb data at large $\eta_l$ and $y_Z$ provide powerful constraints on the quark PDFs at high $x$. The LHCb $Z$ and $W$ production measurements are particularly useful for constraining the independently parameterized charm quark PDF in the NNPDF3.1 fit. In this fit, LHCb data provides one of the primary constraints on the charm quark PDF at $x > 0.1$.

## 3 Forward $W$ + jet and $Z$ + jet production

Measurements of $W$ + jet and $Z$ + jet production probe a larger kinematic region than that probed by inclusive electroweak boson production. In particular, LHCb measurements of $W$ + jet and $Z$ + jet could provide sensitivity to quark PDFs at $x \gtrsim 0.5$ [14]. Interpretation of these measurements is complicated by large factorization and renormalization scale uncertainties. As a result, $W$ + jet and $Z$ + jet production results can be expressed in terms of ratios and asymmetries given by

$$R_{XY} = \frac{\sigma(Xj)}{\sigma(Yj)}, \tag{1}$$

$$A(Wj) = \frac{\sigma(W^+j) - \sigma(W^-j)}{\sigma(W^+j) + \sigma(W^-j)}. \tag{2}$$

These quantities result in the cancellation of many experimental and theoretical systematic uncertainties.

LHCb has measured $W$ +jet and $Z$ +jet production at $\sqrt{s} = 8\,\text{TeV}$ [15]. Both measurements are performed using the muon channels, requiring $p_T^\mu > 20\,\text{GeV}$ and $2.0 < \eta^\mu < 4.5$. Jets are reconstructed using the anti-$k_T$ algorithm with radius parameter $R = 0.5$ and must have $2.2 < \eta^j < 4.2$. The $W$ + jet yield is extracted using a fit to the distribution of the muon isolation variable $p_T^\mu/p_T^{\mu\text{-}jet}$, where $p_T^{\mu\text{-}jet}$ is the transverse momentum of the reconstructed jet containing the muon. A summary of results is shown in Fig. 2.

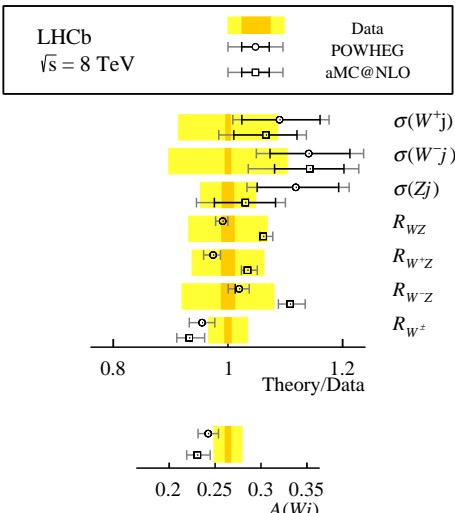

Figure 2: Results of $W$ + jet and $Z$ + jet measurements at $\sqrt{s} = 8\,\text{TeV}$ [15]. The measurements are shown as bands, with the inner band showing the statistical uncertainty and the outer band showing the total uncertainty. The theoretical predictions are shown as points with errorbars, with the inner error bar showing the scale uncertainty and the outer errorbar showing the total uncertainty. The predictions are calculated using the NNPDF3.0 PDF set [16]

## 4 Heavy flavor jets

LHCb's excellent vertex resolution allows for the identification of heavy flavor jets using displaced secondary vertices [17]. Heavy flavor jets are identified by the presence of a displaced secondary vertex. The $b$- and $c$-jet yields are extracted using two boosted decision tree (BDT) classifiers. One BDT is designed to separate heavy and light flavor jets ($BDT_{bc|udsg}$) and the other is designed to separate $b$- and $c$-quark jets ($BDT_{b|c}$). The resulting tagging algorithm identifies $b$ ($c$) jets with 65% (25%) efficiency with a 0.3% light parton mistag probability. The LHCb heavy flavor tagging algorithm has been used to measure $W + c$-jet and $W + b$-jet production at $\sqrt{s} = 7$ and $8\,\text{TeV}$ [18]. $W$ +jet candidates are selected as described in Section 3, and $b/c$-jet yields are extracted using 2D fits to the $BDT_{bc|udsg}$ vs. $BDT_{b|c}$ distributions in bins of the muon isolation. The $W + b/c$ yields are then extracted using template fits to the muon isolation.

Comparisons of LHCb results and theory predictions are shown in Fig. 3. Identifying the jet flavor in $W$ + jet production provides information about the initial parton flavor. $W + c$ is sensitive to the $s$ and $\bar{s}$ PDFs via the process $gs \to Wc$, while $W + b$ probes the $b$ and $\bar{b}$ PDFs via $qb \to Wbq'$. The measured asymmetry $A(Wc)$ disagrees with the NLO QCD calculation by about $2\sigma$. This tension could point to an asymmetry between the $s$ and $\bar{s}$ PDFs.

The LHCb jet tagging algorithm has also been used to measure top-quark production. Most recently, LHCb has measured $t\bar{t}$ production at $\sqrt{s} = 13\,\text{TeV}$ [21]. This measurement uses the high purity $\mu + e + b$-jet final state. Top quark production at LHCb provides information about the gluon PDF at high $x$ and high $Q^2$.

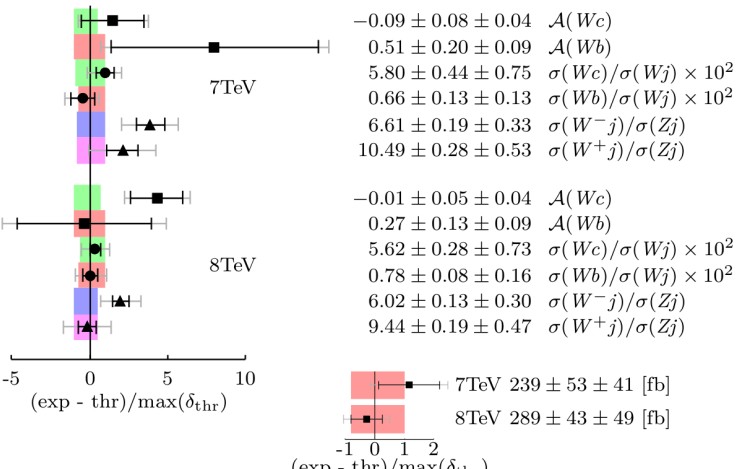

Figure 3: Comparison of $W + b/c$ results and theoretical predicitons [18]. The theory uncertainties are given by the colored bands, while the measurements are shown by points with errorbars. The inner error bar shows the statistical uncertainty, while the outer errorbar shows the total uncertainty. The theoretical predictions were calculated at NLO using MCFM [19] and the CT10 NLO PDF set [20]. The theory uncertainty includes contributions from PDF, scale, and strong coupling uncertainties with the PDF uncertainty dominating.

## 5 Intrinsic charm

Most PDF fits assume that the charm quark PDF is generated perturbatively for $Q^2 > m_c^2$, where $m_c$ is the charm quark pole mass. Charm content in the proton may also arise from an "intrinsic" $|uudc\bar{c}\rangle$ component of the proton wavefunction. The presence of intrinsic charm (IC) implies

$$\langle x \rangle_{\text{IC}} \equiv \int_0^1 xc(x, Q^2 = m_c^2)\mathrm{d}x > 0. \tag{3}$$

Light front QCD calculations predict a valence-like IC contribution to the charm quark PDF [22]. Valence-like IC would result in an increase in the charm quark PDF at high $x$. The NNPDF3.1 PDF set is the first general purpose PDF set to allow for IC, and favors a small valence-like IC contribution with about $1\sigma$ significance, [12].

LHCb has measured charm hadron production in fixed target $p$He and $p$Ar collisions at $\sqrt{s_{\text{NN}}} = 86.6$ and $110.4$ GeV, respectively [23]. These measurements provide sensitivity to the charm quark PDF at high $x$ and low $Q^2$. The measured $D^0$ rapidity distributions are shown in Fig. 4. The results show no evidence for significant intrinsic charm. However, low-$Q^2$ fixed target measurements are difficult to interpret and are usually omitted from PDF fits. Alternatively, $Z + c$ production at LHCb would probe the charm quark PDF in the valence region at high $Q^2$, providing a clean probe of intrinsic charm [24]. A study of $Z + c$ production using Run 2 LHCb data is in progress and should provide sensitivity to valence-like IC with $\langle x \rangle_{\text{IC}}$ as small as about 1%. The same measurement using Run 3 data is expected to be sensitive to $\langle x \rangle_{\text{IC}}$ down to 0.3%.

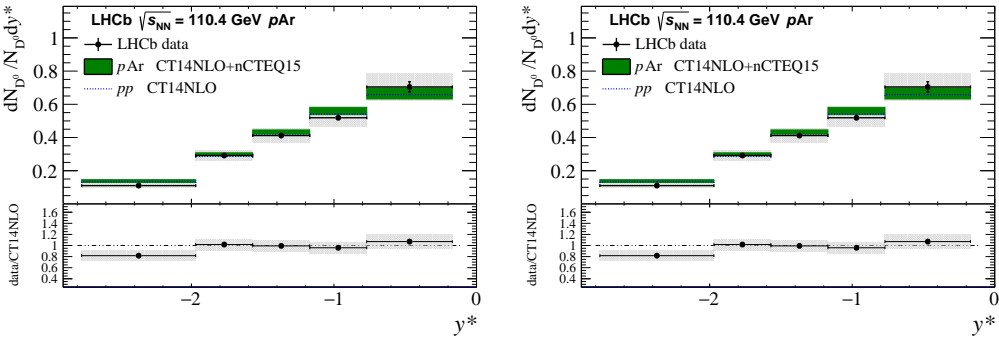

Figure 4: Measured $D^0$ rapidity distributions in fixed target $p$He (left) and $p$Ar collisions [23]. The error bars show the statistical and uncorrelated systematic uncertainties, and the gray shaded region shows the correlated systematic uncertainty.

# 6 Conclusion

LHCb measurements provide significant constraints on proton PDFs at high $x$ in state-of-the-art PDF fits. In addition, LHCb has demonstrated the ability to measure heavy flavor jet production in the forward region. This capability has allowed LHCb to make measurements sensitive to heavy quark PDFs and will allow LHCb probe IC in proton at the level of 1% with Run 2 data and 0.3% in Run 3.

**Funding information** The author is supported by the U.S. National Science Foundation.

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
