# Peer review of "Probing proton PDFs at high x with LHCb"

_SciPost Physics Proceedings, doi:SciPost Phys. Proc. 8, 093 (2022)_

## Round 1 · Referee Report · Anonymous (Referee 1) · 2021-8-9

Strengths

Presents recent LHCb data which can constrain PDFs

Weaknesses

Does not really demonstrate these constraints

Report

This is a short conference report on the impact of LHCb data on high-x PDFs. Whereas it is undoubtedly true that these data can have impact this is only explicitly demonstrated in Fig 1. The rest of the time it is asserted with reference to figures which do not actually show this. In the case of top data there is not even a figure.
I have suggested that information could be added to Figs 2 and 3 below. I have also pointed out that PDFs for intrinsic charm have appeared long before the recent NNPF3.1. If these matter are addressed it can be pulished.

Requested changes

Both Fig 2 and Fig 3 present cross section data compared to theory with minimal information as to what the theory is (particularly for Fig 3). As a minimum the PDF used should be specified and its uncertainties quantified, better still predictions using different PDFs should e compared. Otherwise on cannot claim that these data have impact on PDFs.
it is also stated that NNPDF3.1 is the first general purpose PDFset to allow for intrinsic charm. This is not strictly true the older CTEQ66c set (~2006) did this.

  • validity: ok
  • significance: good
  • originality: high
  • clarity: good
  • formatting: good
  • grammar: good

Author:  Thomas Boettcher  on 2021-09-14  [id 1759]

(in reply to Report 1 on 2021-08-09)

Thank you for the helpful comments. Additional information about the theory calculations has been added in the captions of Figures 2 and 3.

I disagree characterizing CTEQ66c as a "general purpose" PDF set. The CTEQ authors do not characterize it as a general purpose PDF set (see for example https://ct.hepforge.org/PDFs/cteq6.html where CTEQ6, 6.1, 6.5, and 6.6 are all explicitly labeled as "general purpose" while 6.5cn and 6.6C are not). In addition, CTEQ6.5cn and CTEQ6.6C were not published with eigenvector PDF sets to allow for uncertainty calculations.

---

## Editorial Decision

published